# Association of Three Different Dietary Approaches to Stop Hypertension Diet Indices with Renal Function in Renal Transplant Recipients

**DOI:** 10.3390/nu15183958

**Published:** 2023-09-13

**Authors:** I-Hsin Lin, Tuyen Van Duong, Yi-Chun Chen, Shih-Wei Nien, I-Hsin Tseng, Yi-Ming Wu, Yang-Jen Chiang, Hsu-Han Wang, Chia-Yu Chiang, Chia-Hui Chiu, Ming-Hsu Wang, Chia-Tzu Chang, Nien-Chieh Yang, Ying-Tsen Lin, Te-Chih Wong

**Affiliations:** 1Department of Medical Nutrition Therapy, Linkou Chang Gung Memorial Hospital, Taoyuan 333, Taiwan; cabbage@cgmh.org.tw (I.-H.L.); nina0904@cgmh.org.tw (S.-W.N.); cathy40422@cgmh.org.tw (I.-H.T.); yiming1023@cgmh.org.tw (Y.-M.W.); 2School of Nutrition and Health Sciences, College of Nutrition, Taipei Medical University, Taipei 110, Taiwan; tvduong@tmu.edu.tw (T.V.D.); yichun@tmu.edu.tw (Y.-C.C.); 3Department of Urology, Linkou Chang Gung Memorial Hospital, Taoyuan 333, Taiwan; zorro@cgmh.org.tw (Y.-J.C.); seanwang@cgmh.org.tw (H.-H.W.); 4Department of Medicine, Chang Gung University, Taoyuan 333, Taiwan; 5Department of Business Administration, College of Management, National Changhua University of Education, Changhua 500, Taiwan; cychiang@cc.ncue.edu.tw; 6Center for General Education, Taipei Medical University, Taipei 110, Taiwan; thera@tmu.edu.tw (C.-H.C.); mattwang@tmu.edu.tw (M.-H.W.); 7Department of Nutrition and Health Sciences, Chinese Culture University, Taipei 111, Taiwan; patrick241224@gmail.com (C.-T.C.); a9221185@ulive.pccu.edu.tw (N.-C.Y.); 8Department of Health Promotion and Health Education, National Taiwan Normal University, Taipei 106, Taiwan; ariel102551@gmail.com

**Keywords:** renal transplant recipients, Dietary Approaches to Stop Hypertension, diet indices, renal function, glomerular filtration rate

## Abstract

Several dietary indices assess the impacts of the Dietary Approaches to Stop Hypertension (DASH) diet on health outcomes. We explored DASH adherence and renal function among 85 Taiwanese renal transplant recipients (RTRs) in a cross-sectional study. Data collection included demographics, routine laboratory data, and 3-day dietary records. Three separate DASH indices, that defined by Camões (based on nine nutrients), that defined by Fung (using seven food groups and sodium), and that modified by Fung (as above but separated for men and women) were used. Renal function was ascertained through the estimated glomerular filtration rate (eGFR) from patients’ medical records. Participants’ mean age was 49.7 ± 12.6 years and eGFR was 54.71 ± 21.48 mL/min/1.73 m^2^. The three established DASH diet indices displayed significant correlations (r = 0.50–0.91) and indicated the nutritional adequacy of the diet. Multiple linear regressions indicated a significant positive association between higher DASH scores for each index and increased eGFR. In addition, RTRs in the highest DASH score tertile had higher eGFR rates than those in the lowest tertile, regardless of confounding variables. Adherence to a DASH-style diet correlated with better renal function among RTRs. Educating RTRs about the DASH diet may prevent graft function deterioration.

## 1. Introduction

Renal transplantation is a cost-effective method to extend the lifespan of patients with end-stage renal disease [1]. However, graft failure continues to be a substantial burden for renal transplant recipients (RTRs). According to the 2016 Organ Procurement and Transplantation Network/Scientific Registry of Transplant Recipients (OPTN/SRTR) report [2], the 10-year all-cause graft failure rate for the 2006 cohort was 51.6%, which was an improvement compared to the rate recorded for the 1998 cohort (57.2%). In the 2021 OPTN/SRTR report [3], living donor RTRs aged ≥65 years had a 5-year graft survival rate of 82.1%, which was somewhat lower than that of those aged 18–34 years (88.6%); in comparison, deceased donor RTRs aged ≥65 years had a 5-year graft survival rate of 68.0%, which was lower than that of those aged 18–34 years (80.7%). A Taiwanese study also indicated that approximately 20–40% of RTRs experienced unfavorable outcomes within 5–10 years after transplantation, and between 2015 and 2019, 104–144 RTRs required dialysis again [4]. Accordingly, an increasing research interest has been devoted to the factors that impact graft survival after renal transplantation, such as dietary factors.

Osté et al. (2017) [5] determined that adhering to the Dietary Approaches to Stop Hypertension (DASH) diet was related to a reduced risk of decline in renal function and all-cause mortality among RTRs in the Netherlands. However, dietary patterns differ significantly across countries, and the DASH diet scores used in the study relied on sex-specific quintile cutoff values from the Nurses’ Health Study [6], which may not be appropriate for other populations (e.g., males, healthy/subhealthy people or residents of Western/Asian countries). Moreover, previous studies have used different approaches to operationalize the DASH dietary pattern into indices, assessing the risk of hypertension [7], coronary heart disease and stroke [6], and colon cancer [8]. These diverse approaches to operationalizing the DASH dietary pattern have not easily confirmed the correlation between the diet and disease.

In the current study, we compared the scores of different predefined algorithms based on DASH diet indices established in the literature and examined their associations with renal function, as determined by the estimated glomerular filtration rate (eGFR), among RTRs in Taiwan, focusing on the same outcome.

## 2. Participants and Methods

### 2.1. Study Design and Setting

All study procedures were approved by the Institutional Review Board of the Chang Gung Medical Foundation. This observational, cross-sectional, single-center study enrolled stable and ambulatory adult RTRs from the Chang Gung Memorial Hospital since September 2016. Patients who received regular follow-ups were invited to participate in the study through advertisements. Well-trained staff collected data including demographic characteristics, anthropometric measurements, laboratory tests, and dietary intake by using standardized procedures, as described previously [9,10,11].

### 2.2. Patient Recruitment

This study included stable RTRs aged >18 years with graft function monitored using urine albumin-to-creatinine ratio (ACR). All patients were required to maintain immunosuppressive therapy, which consisted of a regimen based on calcineurin inhibitors (CNI), antimetabolites, and steroids. We excluded patients who had had a >3 kg change in body weight, acute rejection, or >25% variation in glomerular filtration rate during the previous 3 months. Ninety RTRs signed the informed consent form, and five patients were excluded due to extreme energy intakes (≤800 or ≥3000 kcal; *n* = 4) and incomplete measurement of body composition (*n* = 1). Figure 1 presents the study flowchart.

### 2.3. Demographics and Anthropometric Measurements

The collected demographic data included sex, age, height (without shoes), and weight (to the nearest tenth of a point; measured twice under fasting conditions, without shoes, and wearing light clothing) measured using electronic scales (HBF-375; Omron Health Care, Tokyo, Japan). These data were obtained from the patients’ medical charts. We computed patient body mass index (BMI) by dividing body weight (in kilograms) by the square of height (in meters).

### 2.4. Laboratory Tests

Blood samples were obtained after at least 8 h of fasting and were analyzed at the clinical laboratories of the Chang Gung Memorial Hospital by using an automated analyzer (Sysmex XN-3000, Kobe, Japan) and standardized methods. The following biochemical parameters were examined: albumin, glucose, glycated hemoglobin A1c (HbA1c), triglycerides, total cholesterol (TC), high-density lipoprotein cholesterol (HDL-C), creatinine, insulin, and high-sensitivity C-reactive protein (hs-CRP). Insulin resistance was determined using the homeostatic model assessment-estimated insulin resistance (HOMA-IR) index, which is calculated as (glucose in mg/dL) × (insulin)/405 [12].

### 2.5. Dietary Data

We used 3-day dietary records (two weekdays and one weekend day) to collect baseline information on dietary patterns among RTRs before their latest visit with a well-trained dietitian. Twenty-four-hour dietary recall was obtained through face-to-face interviews to validate the dietary records. All dietary data were converted into daily energy and nutrient intakes by using nutrient analysis software (COFIT Pro, Version 1.0.0; Cofit HealthCare, Taipei, Taiwan) [13] using the Taiwan Food Composition Table provided by the Ministry of Health and Welfare (MOHW) [14]. DASH index scores were generated using predefined algorithms described previously [6,7]. Table 1 summarizes the scoring standards and points used in these algorithms.

#### 2.5.1. Camões’ DASH Diet Index

The DASH index developed by Camões et al. [7] evaluates adherence to the DASH diet by assessing nine target nutrient values, which are expected to be higher (protein, fiber, calcium, potassium, and magnesium) or lower (total fat, saturated fat, cholesterol, and sodium) with greater adherence to the DASH diet. Additionally, this method establishes absolute targets based on a 1000 kcal diet for both men and women. The scores are assigned for each component as follows: meeting the target, 1 point; meeting an intermediate goal, 0.5 points; and not meeting the proposed intake goal, 0 points. The total score ranges from 0 to 9.

#### 2.5.2. Fung’s DASH Diet Index

The DASH diet index developed by Fung et al. [6] consists of seven food groups (whole grains, vegetables, fruits, legumes and nuts, low-fat dairy products, red and processed meats, and sugar-sweetened beverages) and one nutrient (sodium). The scoring system is based on quintile rankings based on the Nurses’ Health Study. For components where a high intake is desired in the DASH diet (whole grains, vegetables, fruit, legumes and nuts, and low-fat dairy products), a score of 1 (lowest quintile) to 5 (highest quintile) is assigned. Conversely, for the low-intake components (red and processed meats, sugar-sweetened beverages, and sodium), the scoring is reversed (5 for the lowest quintile and 1 for the highest quintile). In Fung’s method, sugar-sweetened beverages include soft drinks and sugar-sweetened fruit drinks. In addition, red and processed meats are calculated by summing the intakes of all types of meat (excluding fish), such as beef, poultry, pork, and lamb. In our study, sodium intake was determined by summing the sodium content in all foods in dietary records using the Taiwan Food Composition Table. Finally, we summarized the component scores to obtain a comprehensive DASH score ranging from 8 (lowest adherence) to 40 (highest adherence).

#### 2.5.3. Modified Fung’s DASH Diet Index

The original Fung’s DASH diet index was based on sex-specific quintile rankings from the Nurses’ Health Study [6], which may not be suitable for men or an Asian population. Therefore, we developed a modified DASH diet index that follows the same seven food groups and one nutrient as the original Fung’s index but categorizes men and women separately into quintiles based on their consumption of each component in our study population. The total scores range from 0 to 40 points.

#### 2.5.4. Assessment of Nutrient Adequacy in the Diet: Nutrient Adequacy Ratio (NAR) and Mean Adequacy Ratio (MAR)

The NAR for 14 nutrients was selected based on the Taiwan MOHW Dietary Reference Intakes (DRIs) for vitamins A, B1, B2, B6, B12, C, and E; folic acid; niacin; calcium; phosphorus; magnesium; zinc; and iron [15]. The NAR was computed for each nutrient by dividing a participant’s daily intake by their age- and sex-specific recommended or reference amounts. The MAR was calculated by summing the 14 NARs (each NAR capped at 1) and dividing the sum by the total number of nutrients [16].

### 2.6. Evaluation of Renal Function

Renal function was determined using the eGFRs retrieved from patients’ medical records, which were calculated using the Modification of Diet in Renal Disease equation: eGFR (mL/min/1.73 m^2^) = 175 × (serum creatinine)^−1.154^ × (age)^−0.203^ × 0.742 (if female) × 1.21 (if African American) [17].

### 2.7. Covariates

We gathered data on medication utilization (such as the usage of steroids) and assessed comorbidities using the Charlson comorbidity index (CCI) [18] from baseline survey data and chart reviews. We also collected information on donor source, transplant history, and immunosuppressant usage through chart review. Blood pressure, including systolic blood pressure (SBP) and diastolic blood pressure (DBP) (averaged over three measurements), was also obtained.

### 2.8. Statistical Analysis

The data were analyzed using SAS software, version 9.4 (SAS Institute, Cary, NC, USA), and a significance level of *p* < 0.05 was used. Normality was evaluated using the Shapiro–Wilk test and Q–Q plot. Descriptive statistics were expressed as mean ± standard deviation unless otherwise stated. Patients were grouped according to the highest and lowest tertiles of total DASH scores for each of the three indices, and the results were compared using appropriate statistical tests, including Student’s t test and the Wilcoxon rank sum test. The association between each DASH score and other variables was estimated using Pearson correlation or Spearman’s rank correlation adjusted for age, sex, and total energy intake. Multiple linear regression analysis was performed to examine the effect of each of the three DASH indices on renal function as indicated by eGFR. Adjustment factors were selected on the basis of their significant impact on eGFR or known prognostic factors. Additionally, we divided the DASH score into tertiles and conducted Dunnett’s t test to compare the impact of the scores in the medium and high tertiles on eGFR, compared with the lowest tertile (control group). The following variable-adjusted models were used: (i) crude model; (ii) model 1: age and sex; (iii) model 2: age, sex, and total energy intake; and (iv) model 2 with BMI, SBP, HbAlc, CCI, and steroid use status.

## 3. Results

Table 2 displays the baseline characteristics of participants in the highest and lowest tertiles of total DASH scores for each of the three indices. The participants’ mean age and posttransplant duration were 49.72 ± 12.60 and 8.83 ± 5.97 years, respectively. Of the included RTRs, 81.2% (*n* = 69) underwent deceased donor transplantation, and a majority of the subjects (*n* = 46) experienced end-stage renal disease (ESRD), primarily due to chronic glomerulonephropathy. Other factors contributing to ESRD included challenges in categorizing patients (*n* = 19), IgA nephropathy (*n* = 5), hypertensive nephrosclerosis (*n* = 5), diabetic nephropathy (*n* = 4), gouty nephropathy (*n* = 3), polycystic kidney disease (*n* = 2), and glomerulosclerosis (*n* = 1). Regarding maintenance immunosuppression regimens, 24.7% (*n* = 21) of RTRs exclusively used CNI therapy, 54 (63.5%) used tacrolimus and 31 (36.5%) used cyclosporine. Additionally, 37.6% (*n* = 32) utilized CNI therapy in combination with a mammalian target of rapamycin (mTOR) inhibitor, while 16.5% (*n* = 14) employed CNI therapy alongside mycophenolate mofetil (MMF). Moreover, 22.4% (*n* = 19) utilized CNI therapy along with both an mTOR inhibitor and MMF. The participants in the highest tertiles of scores across all indices were more likely to have lower height, creatinine levels, total fat and saturated fatty acid, sodium, red and processed meat intake, and calories from fat and saturated fatty acid, as well as higher dietary fiber, vegetable and fruit, whole grain, and cholesterol intake and total calories from carbohydrates. Furthermore, RTRs in the tertile with the highest DASH score had higher eGFRs.

Table 3 presents the correlation coefficients between the total DASH scores for each of the three indices and the NARs of 14 nutrients. Notably, a significant positive correlation was found between each of the three indices and most of the NARs and the MAR.

Table 4 displays the correlations between the total scores for each DASH index. Correlation coefficients varied between 0.50 to 0.91, with the strongest correlation coefficient being found between the Fung’s and modified Fung’s indices (r = 0.91; *p* < 0.0001) and the weakest between Camões’ and Fung’s DASH indices (r = 0.50; *p* < 0.0001).

Multiple linear regressions, which analyzed the total DASH scores for each of the three indices as continuous variables, revealed that a higher DASH score was linked to a higher eGFR when adjusted for age, sex, and total energy intakes. The association remained consistent even after adjustment for BMI, SBP, comorbidities, and steroid use (Table 5).

We also investigated the association between DASH score tertiles and eGFR (Table 6). We observed that the highest scores in all DASH indices were significantly associated with better renal function compared with the lowest scores, regardless of age or sex. The regression coefficients were as follows: Camões’, 18.717 (95% CI: 4.744–32.689; *p* = 0.007); Fung’s, 14.184 (95% CI: 0.372–27.996; *p* = 0.043); and modified Fung’s, 16.113 (95% CI: 2.718–29.509; *p* = 0.016). After adjustment for transplant characteristics and parameters related to renal function, the association remained significant for Camões’ (regression coefficient: 17.418; 95% CI: 4.484–30.352; *p* = 0.007) and modified Fung’s (regression coefficient: 13.708; 95% CI: 1.003–26.414; *p* = 0.033) but only marginally significant for Fung’s (regression coefficient: 12.124; 95% CI: −1.126 to 25.375; *p* = 0.077).

## 4. Discussion

In this cross-sectional study investigating the relationship between three established DASH diet indices and renal function, RTRs who adhered more closely to a DASH-style diet exhibited better renal function. Furthermore, the results suggest that the three established DASH diet indices are interrelated and indicate the nutritional adequacy of the diets (Table 3 and Table 4). The findings also demonstrated that variances in the operationalization of the DASH dietary pattern (e.g., components or scoring criteria) impact the DASH indices’ capability to predict renal function. Our findings reveal a direct or specific effect of dietary quality in protecting against graft failure management in RTRs.

The DASH diet is characterized by its focus on consuming generous amounts of whole grains, vegetables, fruits, low-fat dairy products, fish, poultry, and nuts, and restricting the intake of red meat, sweets, and sugary beverages. These dietary recommendations closely align with the guidelines provided by the National Kidney Foundation (NKF), which promotes a nutritious and balanced diet for RTRs that includes various fresh vegetables and fruits, lean meats, reduced-fat dairy products, and high-fiber and low-salt content in their meal plans [19]. However, limited studies have examined the well-established recommendation of healthy eating patterns for RTRs, particularly in Taiwan. Given that diet plays a crucial role as a modifiable risk factor for chronic diseases, our study provides evidence supporting the adoption of an overall healthy dietary pattern, such as the DASH diet, as a valuable strategy to mitigate the burden of renal function decline in RTRs.

In the present study, we observed a positive association between the DASH diet and renal function, which aligns with previous research findings. Osté et al. (2018) reported a significantly lower risk of renal function decline in stable RTRs in the Netherlands with higher DASH scores [5]. Notably, the DASH diet was associated not only with a reduced risk of renal failure in patients with kidney disease but also with a lower likelihood of chronic kidney disease (CKD) in elderly Korean adults [20]. Lin et al. (2011) recently discovered a significant negative correlation between adherence to the DASH diet and eGFR decline in female participants of the Nurses’ Health Study [21]. According to our review of the relevant literature, numerous studies have examined the association between DASH and renal function; however, few studies have compared established DASH diet indices within the same study, particularly in RTRs.

The NKF acknowledges that the DASH diet is a recognized treatment for hypertension, heart disease, and kidney disease, and it can effectively slow down the progression of kidney diseases [22]. The mechanism by which the DASH diet can protect renal function remains uncertain. We observed a significantly higher intake of vegetables and fruits in the highest tertile of the three established DASH diet indices than in the lowest tertile. In a prospective cohort study from South Korea, Jhee et al. (2019) observed that a diet rich in vegetables and fruits was associated with a reduced risk of incident CKD and proteinuria in participants with normal kidney function at baseline [23]. Plant-based foods, which are an integral part of the DASH diet, are known to be a good source of phytochemicals with anti-inflammatory properties and dietary fiber. Xu et al. (2014) concluded that higher dietary fiber consumption was associated with better renal function in community-dwelling older men from Sweden [24]. This may be attributable to a reduction in acid load. An observational study by Toba et al. (2019) determined that patients with CKD with higher dietary acid load had lower intakes of fruits and vegetables and experienced a greater decline in eGFR [25]. A high dietary acid load can increase metabolic acidosis and contribute to kidney injury through various mechanisms. These mechanisms include increased endothelin-1 levels, which trigger aldosterone production by activating the renin–angiotensin–aldosterone system [26]. Additionally, it can increase ammonium concentration, likely resulting in kidney tubular injury, endothelial dysfunction, and inflammation [27].

Participants in the highest tertile of the three established DASH diet indices had significantly lower intakes of sodium, red meat, and processed meats than those in the lowest tertile. A diet high in sodium or animal protein raises CKD risk. Jardine et al. (2019) evaluated the effect of a dietary salt reduction program on albuminuria in nearly 2000 community-dwelling adults [28]. They discovered that participants in the sodium reduction program had a significantly lower urinary ACR and lower odds of albuminuria, indicating better kidney function. This improvement may have been influenced by factors related to blood pressure. Moreover, studies have demonstrated an association between red meat and processed meats, which are animal protein sources, and increased acid load related to CKD progression [27].

We also observed positive correlations between the three established DASH diet indices and an adequate intake of most micronutrients, suggesting that adherence to the DASH diet reduces the risk of inadequate intake of micronutrients. Additionally, significant positive relationships were identified between the total score calculated from the components of Fung’s DASH index and the MAR, along with marginally significant positive relationships observed in Camões’ DASH index and the MAR. We used Taiwanese DRIs for 14 nutrients as reference values to calculate the NAR. Our findings offer detailed information on nutrient intake and diet quality, demonstrating that adherence to the DASH diet, as reflected by the three established DASH diet indices, is associated with an adequate nutrient intake. However, no significant relationship was observed between Fung’s DASH diet index and the adequate intake of vitamins B1 and B12. Similarly, the association between Camões’ DASH diet index and the adequate intake of several nutrients was either absent or weak. The reasons for these results remain unclear, and future studies should assess dietary quality to consider multiple dimensions simultaneously, such as by incorporating different diet indices and evaluating nutrient adequacy.

To the best of our knowledge, ours is the first study to investigate the association between various established DASH diet indices and renal function in RTRs in an Asian country. However, this study has some limitations. First, because of this study’s cross-sectional design, causality cannot be inferred. Few follow-up studies have investigated the association between the dietary patterns and prognoses of RTRs. Tracking changes in eGFR over an extended period would have provided valuable insights into the effect of diet among RTRs. Further well-designed prospective studies and controlled trials are required to assess whether our findings can be extrapolated. Second, we only considered the sodium content of foods based on dietary records. Quantifying sodium intake is challenging due to the significant variation in salt content between meals consumed outside and those prepared at home. Collecting 24 h urine samples to estimate dietary salt intake is highly recommended. Third, the statistical power of our study may have been constrained due to its relatively small sample size. An important finding of this study is that the modified Fung’s DASH index, which was tailored to our population’s dietary intake, had a stronger predictive impact on renal function in our population. This finding highlights how variations in index composition and scoring algorithms can contribute to inconsistent findings. Future studies must consider the characteristics of the study population and the methods used for assessing dietary intake when designing studies in this field. Finally, although we took into consideration and incorporated potential transplant-related confounders in our multivariable models, it is also important to note that individuals who adhere to a healthy diet, such as the DASH diet, may also be more compliant with other recommendations. Unknown confounders, such as adherence to and the metabolic effects of immunosuppressive medications, the specific primary kidney condition that led to end-stage kidney disease, and considerations like the body composition, exercise habits, and physical function of RTRs, need to be addressed in future studies.

In conclusion, our findings indicate that higher adherence to the DASH dietary pattern was associated with renal function preservation among RTRs, as indicated by eGFR. Although we did not evaluate which DASH diet index was superior, our data provide insights into the variations in components or scoring criteria among different indices and their impact on the observed associations. Promoting education to RTRs on what constitutes a healthy and balanced diet, such as the DASH diet, along with dietary consultations, may present a novel strategy to prevent adverse outcomes.

## Figures and Tables

**Figure 1 nutrients-15-03958-f001:**
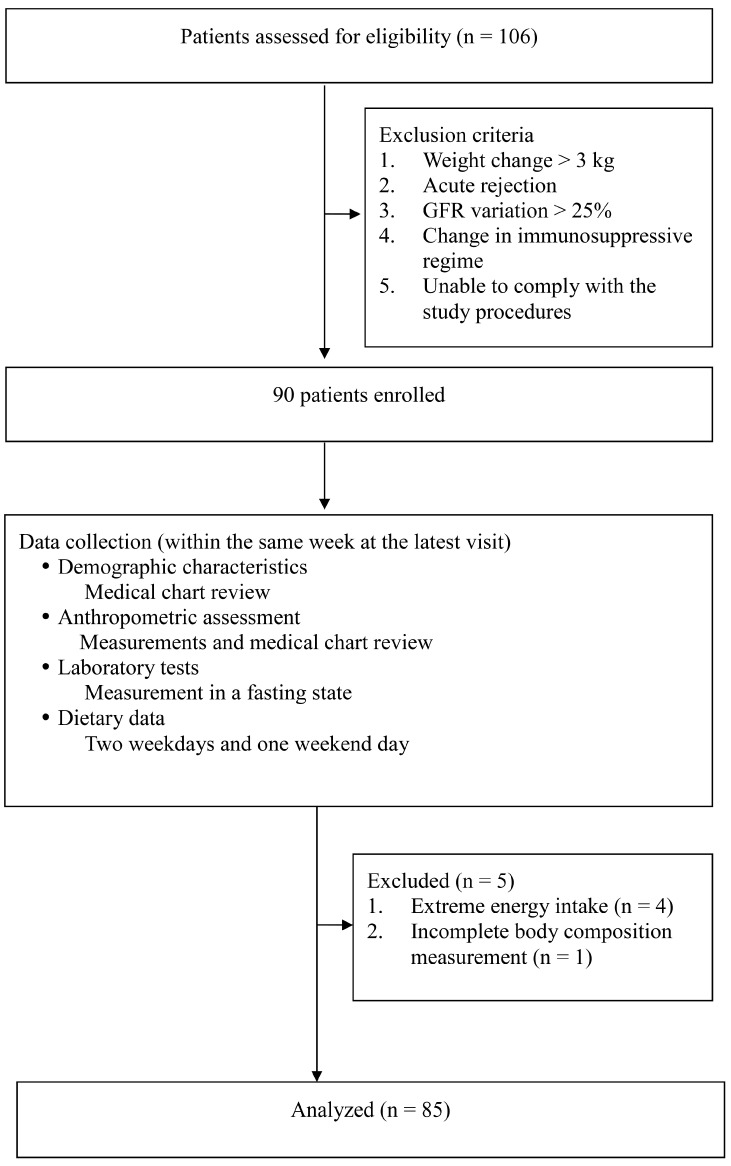
Patient enrollment flowchart. Abbreviations: GFR, glomerular filtration rate.

**Table 1 nutrients-15-03958-t001:** Scoring criteria for each DASH diet index.

Components	Camões’ DASH Index	Fung’s DASH Index	Modified Fung’s DASH Index	
			Male	Female
**Dietary components for which meeting target intake receives higher score **				
Protein, % energy	Higher than 18%	—	—	—
Fiber, g/day	Higher than 14.8 ^1^	—	—	—
Calcium, mg/day	Higher than 590 ^1^	—	—	—
Magnesium, mg/day	Higher than 238 ^1^	—	—	—
Potassium, mg/day	Higher than 2238 ^1^	—	—	—
Fat, % energy	Lower than 27%	—	—	—
Saturated fat, % energy	Lower than 6%	—	—	—
Cholesterol, mg/day	Lower than 71.4 ^1^	—	—	—
Sodium, mg/day	Lower than 1143 ^1^	1041	242.6	366.3
Fruits, servings/day	—	4.1	4.7	3.5
Vegetables, servings/day	—	4.6	6.3	4.1
Legumes and nuts, servings/day	—	1.5	3.8	3.9
Whole grains, servings/day	—	2.4	11.7	10.5
Low-fat dairy products, servings/day	—	2.3	1.0	0.7
Red and processed meats, servings/day	—	0.4	0.0	0.4
Sugar-sweetened beverages, servings/day	—	0	0.0	0.0
**Dietary components for which meeting target intake receives lower score **				
Protein, % energy	Lower than 16.5%	—	—	—
Fiber, g/day	Lower than 9.5 ^1^	—	—	—
Calcium, mg/day	Lower than 402 ^1^	—	—	—
Magnesium, mg/day	Lower than 158 ^1^	—	—	—
Potassium, mg/day	Lower than 1534 ^1^	—	—	—
Fat, % energy	Higher than 32%	—	—	—
Saturated fat, % energy	Higher than 11%	—	—	—
Cholesterol, mg/day	Higher than 107.1 ^1^	—	—	—
Sodium, mg/day	Higher than 1286 ^1^	2676	5503.6	1968.4
Fruits, servings/day	—	0.7	0.0	0.0
Vegetables, servings/day	—	1.1	1.0	0.5
Legumes and nuts, servings/day	—	0.3	0.0	0.0
Whole grains, servings/day	—	0.1	0.0	0.0
Low-fat dairy products, servings/day	—	0.1	0.0	0.0
Red and processed meats, servings/day	—	1.8	6.3	5.4
Sugar-sweetened beverages, servings/day	—	1.2	4.0	1.8

Abbreviations: DASH, Dietary Approaches to Stop Hypertension. ^1^ Per 1000 kcal.

**Table 2 nutrients-15-03958-t002:** Baseline characteristics of demographic, anthropometric, clinical, and nutritional data among renal transplant recipients (*n* = 85) ^1,2^.

	All	Camões’ DASH Index		Fung’s DASH Index		Modified Fung’s DASH Index	
		T1: 0.5–1.0	T3: 2.5–5.0	T1: 3.53–14.66	T3: 18.55–33.45	T1: 7.18–14.2	T3: 18.01–29.02
Number	85	19	29	28	28	28	28
**Demographics **							
Age, year	49.72 ± 12.60	46.79 ± 10.93	51.66 ± 13.71	43.43 ± 10.67	54.93 ± 13.72 *	43.93 ± 10.65	53.82 ± 13.90 *
Male/female	45/40	15/4	13/16	18/10	12/16	18/10	10/17
Renal transplant time, year	8.83 ± 5.97	9.15 ± 5.89	9.37 ± 6.00	8.36 ± 5.54	9.87 ± 6.08	8.31 ± 5.39	9.94 ± 6.27
Tacrolimus/cyclosporine used	54/31	12/7	16/13	25/3	13/15	24/4	17/11
Deceased/living donors	69/16	13/6	24/5	20/8	25/3	20/8	25/3
**Anthropometry **							
Height, cm	161.39 ± 8.61	164.50 ± 8.87	159.47 ± 7.74 *	164.31 ± 9.97	159.22 ± 7.84 *	164.53 ± 10.02	158.63 ± 7.83 *
Body weight, kg	62.88 ± 13.26	69.33 ± 13.63	61.97 ± 11.92	65.34 ± 16.84	60.81 ± 9.32	67.07 ± 16.14	60.06 ± 9.79
Body mass index, kg/m^2^	24.00 ± 3.83	25.53 ± 4.05	24.26 ± 3.57	23.92 ± 4.47	23.94 ± 2.99	24.56 ± 4.46	23.86 ± 3.50
**Laboratory **							
Albumin, g/dL	4.34 ± 0.30	4.20 ± 0.34	4.34 ± 0.27	4.40 ± 0.29	4.32 ± 0.23	4.42 ± 0.29	4.33 ± 0.23
BUN, mg/dL	24.05 ± 11.59	25.79 ± 14.09	20.43 ± 7.07	26.16 ± 12.83	23.3 ± 12.80	27.33 ± 13.14	21.44 ± 11.01
Creatinine, mg/dL	1.43 ± 0.76	1.72 ± 1.13	1.14 ± 0.47 *	1.62 ± 0.99	1.22 ± 0.59 *	1.69 ± 1.00	1.13 ± 0.47 *
TC, mg/dL	208.20 ± 45.34	213.05 ± 48.79	205.24 ± 39.73	211.61 ± 45.96	206.29 ± 48.10	211.00 ± 47.36	200.36 ± 42.05
Triglycerides, mg/dL	157.92 ± 122.19	134.21 ± 40.25	163.48 ± 115.31	139.14 ± 99.56	155.71 ± 67.28	164.64 ± 167.31	138.64 ± 60.57
HDL-C, mg/dL	52.25 ± 17.79	49.74 ± 16.11	52.79 ± 17.93	55.89 ± 19.93	49.89 ± 13.83	54.96 ± 19.62	53.00 ± 14.03
HbA1c, %	6.06 ± 1.01	5.93 ± 0.46	5.99 ± 1.24	5.84 ± 0.63	6.25 ± 1.48	5.92 ± 0.66	6.00 ± 1.28
Insulin, U/mL	8.56 ± 13.04	13.35 ± 26.66	7.59 ± 3.77	7.50 ± 4.46	7.68 ± 3.48	7.92 ± 4.52	7.35 ± 3.59
hs-CRP, mg/dL	5.16 ± 12.20	4.04 ± 4.25	4.45 ± 6.75	4.68 ± 6.10	3.71 ± 5.11	4.49 ± 6.18	3.87 ± 5.06
**Dietary intake**							
Energy, kcal/day	1872.58 ± 377.80	1967.65 ± 410.22	1823.07 ± 429.70	1969.34 ± 345.15	1717.46 ± 367.19 *	1978.72 ± 327.91	1774.18 ± 362.30 *
Carbohydrate, g/day	207.22 ± 47.34	200.46 ± 47.48	216.29 ± 55.92	204.63 ± 45.95	196.09 ± 40.46	205.88 ± 47.54	205.29 ± 40.00
Carbohydrate, % energy	44.53 ± 6.46	40.92 ± 5.90	47.89 ± 7.56 *	41.58 ± 5.85	46.10 ± 5.94 *	41.59 ± 6.32	46.69 ± 5.85 *
Protein, g/day	67.39 ± 14.06	71.25 ± 15.38	65.73 ± 13.67	68.29 ± 13.47	64.53 ± 14.44	69.37 ± 11.74	66.84 ± 13.50
Protein, % energy	14.46 ± 1.76	14.49 ± 1.10	14.65 ± 2.31	13.87 ± 1.51	15.10 ± 2.11 *	14.07 ± 1.39	15.19 ± 2.06 *
Fat, g/day	84.89 ± 22.41	97.51 ± 24.61	75.68 ± 22.63 *	92.18 ± 20.39	76.87 ± 24.42 *	93.27 ± 17.89	78.58 ± 23.66 *
Fat, % energy	40.55 ± 5.71	44.48 ± 5.52	37.10 ± 5.55 *	42.04 ± 5.27	39.72 ± 6.27 *	42.55 ± 5.24	39.36 ± 5.72 *
SFA, g/day	19.47 ± 7.34	27.45 ± 6.79	14.29 ± 5.55 *	22.05 ± 6.75	16.57 ± 7.27 *	21.95 ± 5.62	16.89 ± 7.36 *
SFA, % energy	9.30 ± 2.72	12.55 ± 1.45	7.04 ± 2.15 *	10.12 ± 2.45	8.46 ± 2.53 *	10.13 ± 2.41	8.37 ± 2.73 *
Cholesterol, mg/day	247.58 ± 127.85	215.43 ± 116.51	262.12 ± 133.54 *	254.51 ± 116.02	277.55 ± 151.52 *	246.71 ± 120.21	291.32 ± 143.01 *
Fiber, g/day	13.16 ± 5.37	11.26 ± 4.35	16.20 ± 6.47 *	10.31 ± 4.36	16.81 ± 5.79 *	10.18 ± 4.51	16.93 ± 5.71 *
Calcium, mg/day	347.6 ± 163.18	370.84 ± 198.15	361.20 ± 171.47	373.61 ± 158.25	312.23 ± 142.65	360.89 ± 158.41	326.6 ± 128.26
Magnesium, mg/day	296.79 ± 600.54	165.16 ± 73.06	409.98 ± 829.99	309.65 ± 610.69	288.95 ± 612.64	308.21 ± 611.22	409.07 ± 845.09
Phosphorous, mg/day	726.98 ± 227.99	649.33 ± 220.55	761.14 ± 241.57	749.16 ± 232.06	734.17 ± 250.33	745.03 ± 243.48	765.76 ± 230.24
Potassium, mg/day	1791.88 ± 634.45	1654.19 ± 696.13	1807.86 ± 634.00	1849.53 ± 697.36	1750.5 ± 606.10	1825.59 ± 712.85	1834.10 ± 556.86
Fruits, servings/day	1.21 ± 1.02	0.95 ± 0.96	1.71 ± 1.06 *	0.81 ± 0.90	1.74 ± 1.11 *	0.71 ± 0.87	1.87 ± 1.05 *
Vegetables, servings/day	2.50 ± 1.05	2.08 ± 0.85	2.89 ± 1.16 *	2.15 ± 0.87	2.94 ± 1.25 *	2.09 ± 0.84	2.94 ± 1.28 *
Legumes and nuts, servings/day	0.95 ± 0.95	0.90 ± 0.86	1.15 ± 1.14	0.60 ± 0.84	1.23 ± 0.90 *	0.53 ± 0.58	1.20 ± 1.08 *
Whole grains, servings/day	1.04 ± 2.32	0.44 ± 1.00	1.95 ± 2.96 *	0.04 ± 0.19	2.78 ± 3.37 *	0.13 ± 0.38	2.72 ± 3.40 *
Low-fat dairy products, servings/day	0.06 ± 0.20	0.02 ± 0.07	0.09 ± 0.26	0.01 ± 0.06	0.09 ± 0.26	0.01 ± 0.06	0.14 ± 0.32 *
Sodium, mg/day	1029.83 ± 687.99	1277.43 ± 1138.09	730.21 ± 282.93 *	1385.31 ± 926.50	750.78 ± 349.69 *	1419.24 ± 911.10	732.88 ± 284.34 *
Red and processed meats, servings/day	2.35 ± 1.30	3.08 ± 1.41	1.95 ± 1.22 *	3.20 ± 1.08	1.28 ± 0.74 *	3.17 ± 1.11	1.42 ± 0.78 *
Sugar-sweetened beverages, servings/day	0.53 ± 0.88	0.49 ± 0.64	0.52 ± 1	1.13 ± 1.10	0.07 ± 0.29 *	1.09 ± 1.13	0.11 ± 0.33 *
**Others **							
eGFR, mL/min/1.73 m^2^	54.71 ± 21.48	46.69 ± 20.74	65.39 ± 19.77 *	48.58 ± 20.07	62.76 ± 22.62 *	49.11 ± 18.98	64.98 ± 20.95 *
SBP, mmHg	133.37 ± 15.98	135.63 ± 14.48	130.34 ± 11.70	132.82 ± 16.41	129.43 ± 13.48	132.39 ± 17.53	127.02 ± 12.44
DBP, mmHg	77.90 ± 11.89	79.39 ± 11.84	77.15 ± 9.10	78.31 ± 11.48	74.98 ± 10.61	77.50 ± 12.94	73.79 ± 11.01
HOMA-IR	2.35 ± 4.96	4.00 ± 10.21	2.03 ± 1.71	1.82 ± 1.41	2.10 ± 1.64	1.96 ± 1.42	1.95 ± 1.66
CCI	2.64 ± 0.83	2.47 ± 0.61	2.66 ± 0.90	2.50 ± 0.79	2.61 ± 0.79	2.57 ± 0.84	2.75 ± 1.00

Abbreviations: DASH, Dietary Approaches to Stop Hypertension; TC, total cholesterol; BUN, blood urea nitrogen; HDL-C, high-density lipoprotein cholesterol; HbAlC, glycated hemoglobin A1c; hs-CRP, high-sensitivity C-reactive protein; SFA, saturated fatty acid; eGFR, estimated glomerular filtration rate; SBP, systolic blood pressure; DBP, diastolic blood pressure; HOMA-IR, homeostatic model assessment-estimated insulin resistance; CCI, Charlson comorbidity index. ^1^ Values are indicated as the mean ± standard deviation or number, as appropriate, corresponding to the highest tertiles (T3) and lowest tertiles (T1) of total DASH scores. ^2^ Statistical analyses were conducted using Student’s *t* test or the Wilcoxon rank sum test, as appropriate. * *p* < 0.05.

**Table 3 nutrients-15-03958-t003:** Correlation between total scores for each DASH index and the adequacy of selected nutrients ^1^.

	Correlation Coefficients of		
	Camões’ DASH Index	Fung’s DASH Index	Modified Fung’s DASH Index
Vitamin A (RE)	0.189	**0.302**	**0.278**
Vitamin B1	−0.145	0.170	0.183
Vitamin B2	−0.093	**0.317**	**0.303**
Vitamin B6	0.187	**0.303**	**0.361**
Vitamin B12	0.119	0.080	0.065
Vitamin C	**0.408**	**0.470**	**0.519**
Vitamin E (α-TE)	0.057	**0.290**	**0.280**
Folic acid	**0.301**	**0.596**	**0.584**
Niacin	0.209 ^$^	**0.384**	**0.428**
Calcium	0.125	**0.457**	**0.401**
Phosphorus	0.060	**0.387**	**0.339**
Magnesium	**0.428**	**0.663**	**0.652**
Zinc	0.104	**0.396**	**0.355**
Iron	0.110	**0.358**	**0.312**
MAR	0.214 ^#^	**0.564**	**0.562**

Abbreviations: RE, retinol equivalent; α-TE, α-tocopherol equivalent. ^1^ Statistical analyses were conducted using Pearson’s correlation or Spearman’s rank correlation adjusted for age, sex, and total energy intake, as appropriate. *p* values < 0.05 are indicated in bold. ^#^
*p* = 0.054, ^$^
*p* = 0.06.

**Table 4 nutrients-15-03958-t004:** Correlation coefficients of total scores for the three DASH diet indices in 85 renal transplant recipients ^1^.

	Camões’ DASH Index	Fung’s DASH Index	Modified Fung’s DASH Index
Camões’ DASH index	1	0.501	0.596
Fung’s DASH index	—	1	0.913
Modified Fung’s DASH index	—	—	1

Abbreviations: DASH, Dietary Approaches to Stop Hypertension. ^1^ Statistical analyses were conducted using Pearson’s or Spearman’s rank correlation adjusted for age, sex, and total energy intake, as appropriate. All *p* < 0.0001.

**Table 5 nutrients-15-03958-t005:** Multiple linear regression analysis using renal function, as represented by eGFR as the dependent variable considering the total scores for the three DASH diet indices in 85 renal transplant recipients ^1^.

	Camões’ DASH Index			Fung’s DASH Index			Modified Fung’s DASH Index		
	β	95% CI	*p*	β	95% CI	*p*	β	95% CI	*p*
Crude	7.40	2.70–12.10	0.002	0.99	0.21–1.77	0.014	1.36	0.37–2.36	0.008
Model 1	7.47	2.57–12.38	0.003	0.98	0.13–1.82	0.024	1.35	0.29–2.42	0.013
Model 2	6.87	1.93–11.80	0.007	0.86	0.01–1.71	0.047	1.19	0.11–2.27	0.032
Model 3	6.04	1.44–10.64	0.011	0.88	0.08–1.67	0.031	1.15	0.14–2.16	0.026

Abbreviations: DASH, Dietary Approaches to Stop Hypertension; eGFR, estimated glomerular filtration rate; CI, confidence interval. ^1^ β refers to the regression coefficient to assess the association of the DASH score on renal function, as represented by eGFR. Model 1: adjustment for age and sex. Model 2: model 1 + adjustment for total energy intake. Model 3: model 2 + adjustment for body mass index, blood pressure, HbAlc, comorbidities assessed using the Charlson comorbidity index, and steroid use status.

**Table 6 nutrients-15-03958-t006:** Association between tertiles of DASH diet index scores and renal function, represented by eGFR as the dependent variable ^1^.

	**Camões’ DASH Index**						
	**T1**	**T2**			**T3**		
		**β**	**95% CI**	** *p* **	**β**	**95% CI**	** *p* **
Crude	1 (reference)	3.767	−8.968–16.503	0.714	18.699	5.381–32.017	0.005
Model 1	1 (reference)	3.968	−9.340–17.276	0.707	18.717	4.744–32.689	0.007
Model 2	1 (reference)	4.219	−8.887–17.325	0.670	18.502	4.743–32.262	0.007
Model 3	1 (reference)	6.220	−6.450–18.890	0.421	17.418	4.484–30.352	0.007
	**Fung’s DASH Index**						
	**T1**	**T2**			**T3**		
		**β**	**95% CI**	** *p* **	**β**	**95% CI**	** *p* **
Crude	1 (reference)	4.285	−8.168–16.738	0.656	14.185	1.623–26.747	0.024
Model 1	1 (reference)	4.278	−8.762–17.317	0.675	14.184	0.372–27.996	0.043
Model 2	1 (reference)	4.127	−8.861–17.115	0.692	12.146	−2.022–26.314	0.102
Model 3	1 (reference)	5.076	−7.153–17.305	0.547	12.124	−1.126–25.375	0.077
	**Modified Fung’s DASH Index**						
	**T1**	**T2**			**T3**		
		**β**	**95% CI**	** *p* **	**β**	**95% CI**	** *p* **
Crude	1 (reference)	1.094	−11.104–13.293	0.971	15.864	3.559–28.168	0.009
Model 1	1 (reference)	0.909	−11.807–13.624	0.981	16.113	2.718–29.509	0.016
Model 2	1 (reference)	−0.381	−13.043–12.282	0.997	14.688	1.337–28.040	0.029
Model 3	1 (reference)	3.187	−9.096–15.471	0.783	13.708	1.003–26.414	0.033

Abbreviations: DASH, Dietary Approaches to Stop Hypertension; CI, confidence interval; eGFR, estimated glomerular filtration rate. ^1^ β refers to the regression coefficient to assess the association between DASH score tertiles and renal function, as represented by eGFR. Model 1: adjustment for age and sex. Model 2: model 1 + adjustment for total energy intake. Model 3: model 2 + adjustment for body mass index, blood pressure, HbAlc, comorbidities assessed using the Charlson comorbidity index, and steroid use.

## Data Availability

The data presented in this study can be obtained by contacting the corresponding author upon request. Ethical restrictions prevent the public availability of the data.

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
