# Peer review of "Association of Three Different Dietary Approaches to Stop Hypertension Diet Indices with Renal Function in Renal Transplant Recipients"

_nutrients, 2023, doi:10.3390/nu15183958_

Round 1

Reviewer 1 Report

1 The authors above the word 'nutrients' in the top left corner have removed the information about what type of article it is. The manuscript should be formatted according to the Nutrients format.  

2. In the abstract and at the end of the introduction there are different? objectives of the study. 

What is the purpose of the study? 

3. section 2.1. the place ena number of the bioethics committee is at the end of the manuscript in the statement section - use the Nutrients template please ..... 

4. figure 1 is of very poor quality and there are formatting marks from MS WORD .... 

5. the authors did not describe the procedure for performing the body composition analysis - in this type of study a lot depends on the preparation of the patient for the study. 

6. Omron HBF-375 is not a certified medical device - I did not find this information on the manufacturer's website or in the instruction manual. The quality and repeatability of the measurement in this case is questionable. 

7. line 94 - should be body weight? not "dividing weight"

8. on what laboratory equipment were the blood tests performed? This information is missing

9. section 2.5.3 no citation: 

The purpose of the study should be more specific or the authors should indicate the endpoint: e.g. renal function by egfr etc. It was only in the conclusion (which is good and supported by the results) that I found out what the endpoint of the study was. 

I am glad that the authors pointed out the importance of dietary consultation - which should be given by a dietician unless there are different regulations in the country. 

minior

Author Response

#Reviewer 1

  1. The authors above the word 'nutrients' in the top left corner have removed the information about what type of article it is. The manuscript should be formatted according to the Nutrients format.

Response: We apologize for the manuscript not adhering to the Nutrients format as intended in the original submission. After a thorough review of the Instructions for Authors, we have substantially revised the text to align with the required formatting.

  1. In the abstract and at the end of the introduction there are different? objectives of the study. What is the purpose of the study?

Response: Thank you for your comprehensive review. We apologize if the original manuscript's description of the study objectives and purpose was unclear. While various dietary indices have been developed to capture the Dietary Approaches to Stop Hypertension (DASH) eating pattern and explore its connections with health outcomes, a direct comparison within renal transplant recipients (RTRs), particularly in Taiwan, has been lacking. In this study, we compared three distinct DASH indices: one defined by Camões (based on nine nutrients), another by Fung (utilizing seven food groups and sodium), and a modified version by Fung (adjusted for men and women). We examined their associations with renal function in RTRs within the same study, with a specific focus on the same outcome. To emphasize the study's objectives and purpose, we have revised the Introduction section (Page 3, lines 76-83).

  1. section 2.1. the place ena number of the bioethics committee is at the end of the manuscript in the statement section - use the Nutrients template please .....

Response: Thank you for providing this direction. We had remove the number of the bioethics committee to the end of the manuscript in the statement section (Page 21, lines 371-373).

  1. figure 1 is of very poor quality and there are formatting marks from MS WORD ....

Response: Thank you for providing this direction. We have adjusted the figure formats to JPEG in accordance with the Nutrients format.

  1. the authors did not describe the procedure for performing the body composition analysis - in this type of study a lot depends on the preparation of the patient for the study.

Response: We apologize if the description of the procedure for measuring anthropometric measurements was unclear in the initial manuscript. In this study, we solely measured body weight as an anthropometric measurement using a reliable weighing scale. The measurement process for evaluating body weight was carried out during the most recent visit. Participants were measured twice under fasting conditions, without shoes, and wearing light clothing, as detailed in the revised manuscript (page 6, lines 109-111). Additionally, we have noted in the limitation section that future research could validate the impact of body composition on the outcome (page 21, lines 346).

  1. Omron HBF-375 is not a certified medical device - I did not find this information on the manufacturer's website or in the instruction manual. The quality and repeatability of the measurement in this case is questionable.

Response: Thank you for providing this direction. In this study, we utilized the Omron HBF-375 to solely assess body weight. Numerous studies have employed this device for measuring body weight. A cross-sectional study conducted by Kamruzzaman et al. (2021) involved the measurement of body weight for 300 participants (Plos One 16.9 (2021): e0257055). Similarly, Ghosh et al. (2023) employed this device to measure the body weight of 650 adults in an epidemiological study (Scientific Reports 13.1 (2023): 4895). For further information, the user manual for the Omron HBF-375 can be accessed through the following website:

https://www.omronhealthcare-ap.com/Content/uploads/products/789b8222779742fe808151a86d9851e4.pdf

  1. line 94 - should be body weight? not "dividing weight"

Response: We apologize if the explanation of body mass index (BMI) was not clear in the initial manuscript. BMI is calculated by dividing weight in kilograms by the square of height in meters (Page 6, lines 112-113).

  1. on what laboratory equipment were the blood tests performed? This information is missing

Response: Thank you for providing this direction. The blood tests were conducted using the Sysmex XN-3000 analyzer (Sysmex, Kobe, Japan). We have incorporated this information into the revised manuscript (page 6, lines 117).

  1. section 2.5.3 no citation:

Response: Thank you for providing this direction. We have included the citation in the revised manuscript (page 9, lines 159-160).

  1. The purpose of the study should be more specific or the authors should indicate the endpoint: e.g. renal function by egfr etc. It was only in the conclusion (which is good and supported by the results) that I found out what the endpoint of the study was.

Response: Thank you for offering these instructions. We have made revisions to the text to emphasize the study objectives and purpose. Additionally, we've indicated the study endpoint as the estimated glomerular filtration rate (eGFR) (Page 3, lines 80-84).

  1. I am glad that the authors pointed out the importance of dietary consultation - which should be given by a dietician unless there are different regulations in the country.

We are grateful for the chance to revise our work and sincerely hope that our revisions meet your approval. We appreciate your valuable time and effort in guiding us to enhance the paper.

Reviewer 2 Report

The abstract needs to be rewritten and conveyed better to the reader. The flow and usage of the information does not read well. The headings in the abstract need to be removed since they do not conform to journal standards. 

The first sentence in the introduction, "Renal transplantation is the most effective method for extending the lifespan of patients with chronic kidney disease (CKD)," is inaccurate and should be revised. 

The second sentence needs a reference. 

Overall, the introduction is weak in conveying why the study needed to be conducted. Also, the rationale for consuming less sodium to address CKD is not entirely true. An individual has elevated sodium levels because they have CKD, not because they have high sodium. 

What equations were employed to obtain eGFR?

Figure 1 is blurring and needs to be adjusted via resolution. 

In Table 2, what does the asterisk indicate where the significant statistical difference was at? Also, what does T1-3 mean? This information needs to be better explained. 

Table 3. Remove the #p = 0.054 and 224 $p = 0.06 from the legends and the table. The results are either significant or they are not. 

I don't see the benefit of having Table 4. The amount of information is minimal and can be described in the text. 

Page 13 is blank and needs to be removed. 

The limitations need a separate paragraph. 

The overall significance of the literature is low in terms of what is known regarding diets and kidney disease. 

English verbiage, sentence structure, and syntax need to be improved. Overall, the quality of the English is ok. 

Author Response

#Reviewer 2

  1. The abstract needs to be rewritten and conveyed better to the reader. The flow and usage of the information does not read well. The headings in the abstract need to be removed since they do not conform to journal standards.

Response: We apologize for the manuscript not adhering to the Nutrients format as intended in the original submission. After a thorough review of the Instructions for Authors, we have substantially revised the text to align with the required formatting.

  1. The first sentence in the introduction, "Renal transplantation is the most effective method for extending the lifespan of patients with chronic kidney disease (CKD)," is inaccurate and should be revised.

Response: Thank you for providing this direction. We have rephrased the sentence as follows: "Renal transplantation is a cost-effective method to extend the lifespan of patients with end-stage renal disease” referencing Abecassis et al., (2008) (Clinical Journal of the American Society of Nephrology: CJASN, vol. 3, 2 (2008): 471-80) (Page 3, lines 59-60).

  1. The second sentence needs a reference.

Response: We apologize if the original manuscript's description was unclear. After a thorough evaluation, we have chosen to delete this sentence to enhance the coherence of the article's flow

  1. Overall, the introduction is weak in conveying why the study needed to be conducted. Also, the rationale for consuming less sodium to address CKD is not entirely true. An individual has elevated sodium levels because they have CKD, not because they have high sodium.

Response: Thank you for your comprehensive review. We apologize if the original manuscript's description of the study objectives and purpose was unclear. While various dietary indices have been developed to capture the Dietary Approaches to Stop Hypertension (DASH) eating pattern and explore its connections with health outcomes, a direct comparison within renal transplant recipients (RTRs), particularly in Taiwan, has been lacking. In this study, we compared three distinct DASH indices: one defined by Camões (based on nine nutrients), another by Fung (utilizing seven food groups and sodium), and a modified version by Fung (adjusted for men and women). We examined their associations with renal function in RTRs within the same study, with a specific focus on the same outcome. To emphasize the study's objectives and purpose, we have revised the Introduction section (Page 3, lines 78-83). Furthermore, we acknowledge your comment regarding individuals having elevated sodium levels due to chronic kidney disease (CKD). However, previous studies have indicated that a high-sodium diet raises the risk of CKD (Page 20, lines 309-316). Notably, many individuals, including CKD patients, consume excessive sodium in their diet. Therefore, it is recommended to consider a moderated restriction of daily sodium intake in the diet.

  1. What equations were employed to obtain eGFR?

Response: We apologize if the equations of estimated glomerular filtration rate (eGFR) was not clear in the initial manuscript. The eGFR was calculated using the Modification of Diet in Renal Disease equation: eGFR (mL/min/1.73 m2) = 175 × (serum creatinine)−1.154 × (age)−0.203 × 0.742 (if female) × 1.21 (if African American), and retrieved from patients’ medical records (Page 9, lines 173-176).

  1. Figure 1 is blurring and needs to be adjusted via resolution.

Response: Thank you for providing this direction. We have adjusted the figure formats to JPEG in accordance with the Nutrients format.

  1. In Table 2, what does the asterisk indicate where the significant statistical difference was at? Also, what does T1-3 mean? This information needs to be better explained.

Response: Thank you for providing this direction. We apologize if the original manuscript's description of the asterisk indicating and the meanings of T1-3 were unclear. The asterisk signifies a significant difference determined using either the Student's t-test or the Wilcoxon rank-sum test. Additionally, we compared the baseline characteristics of participants in the highest tertiles (T3) and lowest tertiles (T1) of total DASH scores for each of the three indices. This information has been integrated into the revised manuscript (page 13, lines 215-217).

  1. Table 3. Remove the #p = 0.054 and $p = 0.06 from the legends and the table. The results are either significant or they are not.

Response: Thank you for providing this direction. We do recognize that p = 0.054 and p = 0.06 do not indicate a statistically significant difference. However, owing to the limited sample size in this study, this outcome holds marginal significance, thus warranting special notation. It may also serve as a valuable reference point for future research endeavors.

  1. I don't see the benefit of having Table 4. The amount of information is minimal and can be described in the text.

Response: We appreciate your guidance. The distinct compositions of the indexes can also impact the outcomes. Camões' DASH diet index incorporates components based on nutrients, whereas Fung's DASH diet index utilizes food groups. The intention of this study was not to establish a superior DASH diet index, but rather to highlight the methodological disparities among these indexes and explore their impact on the observed relationships. Should a substantial correlation emerge between these two evaluation methods, simplifying the evaluation process might be conceivable, thereby enhancing research analysis efficiency for large-scale studies in the future.

  1. Page 13 is blank and needs to be removed.

Response: We apologize for the presence of the blank page in the original submission. We have extensively revised the text to ensure alignment with the required formatting.

  1. The limitations need a separate paragraph.

Response: We appreciate your guidance. We adhering to the Nutrients format f the Instructions for Authors, and the limitation section is separate paragraph in line 329-346 (Page 21)

  1. The overall significance of the literature is low in terms of what is known regarding diets and kidney disease.

We are grateful for the chance to revise our work and sincerely hope that our revisions meet your approval. We appreciate your valuable time and effort in guiding us to enhance the paper.

Reviewer 3 Report

The study aimed to assess the effectiveness of a dietary approach on kidney transplant patients. The results indicated a positive connection between dietary style and improved kidney function. However, a significant aspect needs consideration in this study as follows:

The primary concern is that the authors only examined the situation at a single point in time, without tracking changes over an extended period. It would have been more valuable to observe variations in eGFR over time, preferably using at least two data points. This approach could provide insights into factors influencing eGFR decline. Considering the purpose of this study analyses only using a single time point data cannot be acceptable.

The explanation of the authors' utilization of a mixed linear model remains unclear to the reviewer. To comprehensively analyze factors tied to eGFR decline, it is essential to provide a clearer understanding of this modeling approach (In general, the authors need at least two time point data of eGFR to analyze the factors associated with the eGFR decline using a mixed linear model...)

Moreover, delving beyond a literature review, the authors should expand the discussion by incorporating their own data. For instance, exploring potential links between dietary patterns and factors like blood pressure or albuminuria might enhance the discussion on the dietary pattern's beneficial mechanisms.

To enhance the study's context, details about the primary kidney disease that led to end-stage kidney disease should be provided.

Additionally, information about the usage of immunosuppressants other than calcineurin inhibitors would be insightful. (This information may support considering the serum concentration of calcineurin inhibitors. Although calcineurin inhibitor is one of the most important factors associated with the decline of eGFR, if the patients are treated with multiple immunosuppressants, the concentration of the need for the calcineurin inhibitor may be decreased).

Author Response

#Reviewer 3

  1. The primary concern is that the authors only examined the situation at a single point in time, without tracking changes over an extended period. It would have been more valuable to observe variations in eGFR over time, preferably using at least two data points. This approach could provide insights into factors influencing eGFR decline. Considering the purpose of this study analyses only using a single time point data cannot be acceptable.

Response: Thank you for conducting a thorough review. Your guidance is greatly appreciated. Tracking changes in estimated glomerular filtration rate (eGFR) over an extended period would have provided valuable insights into the effect of diet among renal transplant recipients (RTRs). However, due to limitations in study funding, our study was designed as cross-sectional. This limitation led us to focus solely on analyzing the relationship between a single eGFR and the DASH dietary score. To ensure the accuracy of our research findings, we made efforts to include potential confounding factors that could affect eGFR in our regression model (Table 5). Additionally, we explored the DASH score using both continuous values and tertiles to determine whether a stronger protective effect on eGFR was consistently associated with higher DASH scores (Table 6). Importantly, one of the inclusion criteria for participant recruitment in this study was to target individuals whose GFR had remained consistent, with no more than a 25% variation in GFR during the previous 3 months (Figure 1). This requirement aims to minimize the potential impact of eGFR fluctuations on the outcomes of this study. We have incorporated your suggestions within the scope of this study and hope that future research will delve deeper into exploring the relationship between the DASH diet and eGFR values at various time points (Page 20, lines 332-333).

  1. The explanation of the authors' utilization of a mixed linear model remains unclear to the reviewer. To comprehensively analyze factors tied to eGFR decline, it is essential to provide a clearer understanding of this modeling approach (In general, the authors need at least two time point data of eGFR to analyze the factors associated with the eGFR decline using a mixed linear model...)

Response: We apologize if the description of the statistical analysis for Table 5 was unclear in the initial manuscript. We categorized the DASH score into tertiles to assess the influence of scores within the medium and high tertiles on eGFR, relative to the lowest tertile (control group). As a result, we employed multiple linear regression, using a single GFR point (as the dependent variable), and the DASH score tertile as the independent variable. We have extensively revised the text.

  1. Moreover, delving beyond a literature review, the authors should expand the discussion by incorporating their own data. For instance, exploring potential links between dietary patterns and factors like blood pressure or albuminuria might enhance the discussion on the dietary pattern's beneficial mechanisms.

Response: We appreciate your guidance. We have expanded our discussions to encompass the connection between adhering to the DASH diet's dietary patterns and their impact on blood pressure and albuminuria. We've explored aspects like the antioxidant properties of various components in the DASH diet, the role of dietary fibers in reducing acid load and mitigating kidney tubular injury, as well as addressing endothelial dysfunction and inflammation (Lines 293-308). Additionally, the influence of lower sodium intake can be linked to factors tied to blood pressure, and the consumption of animal protein can contribute to an acid load that's associated with the progression of CKD (Lines 309-316).

  1. To enhance the study's context, details about the primary kidney disease that led to end-stage kidney disease should be provided.

Response: We apologize for not furnishing details regarding the primary kidney disease that resulted in end-stage kidney disease within our data collection. Your suggestions have been integrated into the limitations section as recommended. (Lines 345-346)

  1. Additionally, information about the usage of immunosuppressants other than calcineurin inhibitors would be insightful. (This information may support considering the serum concentration of calcineurin inhibitors. Although calcineurin inhibitor is one of the most important factors associated with the decline of eGFR, if the patients are treated with multiple immunosuppressants, the concentration of the need for the calcineurin inhibitor may be decreased).

Response: Your guidance is highly valued. However, it's important to note that one of the criteria for including participants in this study was the requirement to maintain immunosuppressive therapy (Figure 1). Monitoring the serum concentration of calcineurin inhibitors could have provided valuable insights. Additionally, we have incorporated your suggestions into the limitations section (Lines 345-346).

We are grateful for the chance to revise our work and sincerely hope that our revisions meet your approval. We appreciate your valuable time and effort in guiding us to enhance the paper.

Round 2

Reviewer 1 Report

ONLY MINIOR CORRECTION:

My last comment: 

  1. Omron HBF-375 is not a certified medical device - I did not find this information on the manufacturer's website or in the instruction manual. The quality and repeatability of the measurement in this case is questionable.

Response: Thank you for providing this direction ( ... ) etc. 

Thank you for pointing out the paper, but it does not change the fact that this device is not medical and the authors made a mistake used to BIA analysis. 

It is acceptable for measuring body weight (only!), but not for performing bioelectrical impedance analysis for scientific purposes. 

The Methods paragraph should be corrected with the sentence "using bioelectrical impedance analysis scales". - as they did not perform the BIA test. 

Therefore, it is sufficient that: the body weight was measured with an scale (OMRON etc)  ....  

minior

Author Response

#Reviewer 1

  1. Thank you for pointing out the paper, but it does not change the fact that this device is not medical and the authors made a mistake used to BIA analysis. It is acceptable for measuring body weight (only!), but not for performing bioelectrical impedance analysis for scientific purposes. The Methods paragraph should be corrected with the sentence "using bioelectrical impedance analysis scales". - as they did not perform the BIA test.  Therefore, it is sufficient that: the body weight was measured with an scale (OMRON etc) 

Response: hank you for providing this direction. We have rephrased the sentence as follows: “weight (to the nearest tenth of a point; measured twice under fasting conditions, without shoes, and wearing light clothing) measured using electronic scales (HBF-375; Omron Health Care, Japan).” (lines 92-94).

    We are grateful for the chance to revise our work and sincerely hope that our revisions meet your approval. We appreciate your valuable time and effort in guiding us to enhance the paper.

Reviewer 2 Report

Thank you for addressing my comments and revision suggestions. 

The authors adequately addressed my concerns. 

Author Response

#Reviewer 2

  1. The authors adequately addressed my concerns. 

Response: We are grateful for the chance to revise our work. We appreciate your valuable time and effort in guiding us to enhance the paper.

Reviewer 3 Report

The authors revised the manuscript based on the reviewer’s comments.

Although queries 2 and 3 have been addressed, queries 1, 4, and 5 have not been well addressed. Because of this, the reliability of the current study is limited.

Unless the authors address queries 1,4 and 5, the authors cannot identify their study purpose.  Therefore, I cannot recommend accepting the current version. In general, information concerning queries 4 and 5 should be able to obtain at least.

Author Response

#Reviewer 3

  1. The authors revised the manuscript based on the reviewer’s comments. Although queries 2 and 3 have been addressed, queries 1, 4, and 5 have not been well addressed. Because of this, the reliability of the current study is limited. Unless the authors address queries 1,4 and 5, the authors cannot identify their study purpose. Therefore, I cannot recommend accepting the current version. In general, information concerning queries 4 and 5 should be able to obtain at least.

Response: Regarding Query 1, due to the cross-sectional design of this study, we apologize once again for presenting results based on a single eGFR value only. We have further elaborated on the valuable suggestions you provided in the limitations section of the revised manuscript (lines 349-351).

Query 4 focused on investigating the causes of end-stage renal disease (ESRD) in the 85 subjects of this study. We dedicated our best efforts to reviewing the patients' case information within the allocated time. A majority of the subjects (n = 46) experienced ESRD primarily due to chronic glomerulonephropathy. Other factors contributing to ESRD included challenges in categorizing patients (n = 19) resulting from transfers from external hospitals, the absence of electronic medical records or diagnostic records, IgA nephropathy (n = 5), hypertensive nephrosclerosis (n = 5), diabetic nephropathy (n = 4), gouty nephropathy (n = 3), polycystic kidney disease (n = 2), and glomerulosclerosis (n = 1). The aforementioned data has been incorporated into the revised manuscript (lines 198-203).

Query 5 investigated information regarding the utilization of immunosuppressants other than calcineurin inhibitors. The results showed that 24.7% (n = 21) of renal transplant recipients (RTRs) exclusively used calcineurin inhibitor (CNI) therapy (cyclosporine-based or tacrolimus FK-based regimen). Additionally, 37.6% (n = 32) utilized CNI therapy in combination with a mammalian target of rapamycin (mTOR) inhibitor, while 16.5% (n = 14) employed CNI therapy alongside mycophenolate mofetil (MMF). Moreover, 22.4% (n = 19) utilized CNI therapy along with both an mTOR inhibitor and MMF.  The aforementioned data has been incorporated into the revised manuscript (lines 203-208).

We are grateful for the chance to revise our work and sincerely hope that our revisions meet your approval. We appreciate your valuable time and effort in guiding us to enhance the paper.